# Antimicrobial Activity of Divaricatic Acid Isolated from the Lichen *Evernia mesomorpha* against Methicillin-Resistant *Staphylococcus aureus*

**DOI:** 10.3390/molecules23123068

**Published:** 2018-11-23

**Authors:** Jong Min Oh, Yi Jeong Kim, Hyo-Seung Gang, Jin Han, Hyung-Ho Ha, Hoon Kim

**Affiliations:** 1Department of Pharmacy, and Research Institute of Life Pharmaceutical Sciences, Sunchon National University, Suncheon 57922, Korea; ddazzo005@naver.com (J.M.O.); anjin5970@naver.com (J.H.); hhha@scnu.ac.kr (H.-H.H.); 2Department of Agricultural Chemistry, Sunchon National University, Suncheon 57922, Korea; dlwjd0608@naver.com (Y.J.K.); tomodo1998@naver.com (H.-S.G.)

**Keywords:** lichen, *Evernia mesomorpha*, divaricatic acid, antimicrobial activity, minimum inhibitory concentration (MIC), methicillin-resistant *Staphylococcus aureus* (MRSA)

## Abstract

One hundred and seventy seven acetone extracts of lichen and 258 ethyl acetate extracts of cultured lichen-forming fungi (LFF) were screened for antimicrobial activity against *Staphylococcus aureus* and *Enterococcus faecium* using a disk diffusion method. Divaricatic acid was isolated from *Evernia mesomorpha* and identified by LC-MS, ^1^H-, ^13^C- and DEPT-NMR. Purified divaricatic acid was effective against Gram + bacteria, such as *Bacillus subtilis*, *Staphylococcus epidermidis*, *Streptococcus mutans*, and *Enterococcus faecium*, with the minimum inhibitory concentration (MIC) values ranging from 7.0 to 64.0 μg/mL, whereas vancomycin was effective in the MICs ranging from 0.78 to 25.0 μg/mL. Interestingly, the antibacterial activity of divaricatic acid was higher than vancomycin against *S. epidermidis* and *E. faecium*, and divaricatic acid was active against *Candida albicans*. In addition, divaricatic acid was active as vancomycin against *S. aureus* (3A048; an MRSA). These results suggested that divaricatic acid is a potential antimicrobial agent for the treatment of MRSA infections.

## 1. Introduction

Lichens are composite living organisms comprised of algae (photobiont) and fungi or cyanobacteria (mycobiont) [1,2], and produce numerous biologically active secondary metabolites with, for example, antiviral, antibiotic, antitumor, or anti-allergenic properties [3]. A huge number of secondary metabolites from more than 20,000 species of lichens have been investigated as potential scaffolds for drug development [2,4,5]. Usnic acid is probably the most investigated compound of these metabolites, and has antibiotic activity and has been used to develop pharmaceutics and cosmetics [6,7,8]. Recently, usnic derivatives were synthesized to enhance its antimicrobial activity [9].

*Staphylococcus aureus* is a Gram-positive bacterium found on skin and mucosal surfaces, and is a normal inhabitant of the healthy human body. However, in some individuals, such as immunocompromised patients, *S. aureus* can be pathogenic and cause a range of conditions from minor skin infections to life-threatening diseases, such as pneumonia, endocarditis, and sepsis [10]. Furthermore, *S. aureus* is related to nosocomial infections and methicillin-resistant *S. aureus* (MRSA) has become a major health problem [11,12]. Recently, antibiotic compounds of new classes were developed by two step or convenient synthesis to be active against MRSA [13,14]. Antibiotics for staphylococci target the cell envelope, ribosomes, and nucleic acids [15], and vancomycin has proven efficacy against MRSA infections and has been administered even other treatments have failed [16,17]. *Enterococcus faecium* is a Gram-positive bacterium found in the human gastrointestinal tract, causes diseases, such as neonatal meningitis or endocarditis, and has recently emerged as a therapeutically challenging pathogen [18,19]. 

Divaricatic acid has been reported in extracts of the lichens *Ramalina hierrensis* [20], *Flavocetraria nivalis* and *Ophioparma ventosa* [21], and *Lepraria* sp. [22], and the cyclohexane extracts four Algerian lichens, such as *Cladonia rangiformis*, *Ramalina farinaceae*, *R. fastigiata*, and *Roccella phycopsis*, were reported to be active against human pathogenic fungi [23]. Divaricatic acid isolated from *Ramalina aspera* was reported to have molluscicidal and antiparasitic activities [24], and that from *Lecanora frustulosa* had minimal inhibition concentration (MIC) values from 0.78 to 12.5 mg/mL [25]. 

In this study, we isolated divaricatic acid as the main constituent from an acetone extract of *Evernia mesomorpha* and investigated its antimicrobial activities, and its activities against MRSA and *Candida albicans*.

## 2. Results and Discussion

### 2.1. Antimicrobial Activity

#### 2.1.1. Screening

Antimicrobial activities of 435 lichens were determined using 1.0 mg of extracts spotted onto paper disks. Based on halo sizes, samples tested were divided into three groups, i.e., low, moderate, and effective). None of the 258 EA extracts of cultured LFF were active against the tested bacteria, *S. aureus* and *E. faecium*. Of the 177 acetone extracts of lichens, 9 and 28 samples were effective against *S. aureus* and *E. faecium*, respectively, and 8 samples of them were effective against both strains. Total 29 samples were selected for further evaluation (Table 1).

#### 2.1.2. Assay of Antimicrobial Activity

In antimicrobial activity testing of these samples, halo sizes for *S. aureus* were smaller than those for *E. faecium* (Figure 1). The tests were carried out with duplicate experiments. Diameter of halo was measured including disk (8 mm). Halo size order of the samples for *S. aureus* was 419 > 407 = 431, and those for *E. faecium* was 449 > 407 > 458 = 517 (Table 1). 

### 2.2. HPLC of the Selected Extracts

Prior to selecting extracts for further study, we searched literatures for information on the components and biological activities of the 34 extracts. Based on halo sizes and literature information, we analyzed 8 extracts by HPLC. Six peaks were observed in sample 458 (*Evernia mesomorpha*) (Figure 2), and other samples produced four to seven peaks (Appendix A). Components were identified by comparing their retention times to standard data (Table 2). Of those compounds identified, usnic acid is well-known to have antimicrobial activity and galbinic acid, atranorin, and chloroatranorin have also been reported to be active [26,27]. We observed divaricatic acid was 17.8 times more abundant than usnic acid in *E. mesomorpha* extract, and that it exhibited high antimicrobial activity in the screening experiment. *Usnea* sp. showed the best antimicrobial activity, however, the major component was usnic acid; sample 458 showed less activity than *Usnea* sp., however, the major component was not well studied in its antimicrobial activity. In view of its antimicrobial activity, novelty, and availability, *E. mesomorpha* and divaricatic acid were selected for further study.

### 2.3. Structural Analysis of the Major Active Compound in E. mesomorpha

In the mass spectrometric analyses, the values of *m*/*z* ratios of the major MS peaks detected in negative ionization mode indicated the formation of [M − H]^−^ ions (C_21_H_23_O_7_-, *m*/*z* = 387) (Appendix A). The ^1^H-NMR data indicated the presence of one carboxyl group, two benzene ring protons, one methoxy group, and two propyl groups (Appendix A). We compared the MS and ^1^H-NMR data obtained with those reported for divaricatic acid [28]. Based on the twenty-one carbon peaks in ^13^C-NMR (Appendix A) and protonated carbon shifts from DEPT (distortionless enhancement by polarization transfer) (Appendix A), the compound was positively identified as divaricatic acid (Figure 3). Purity of the divaricatic acid by LC was 97.1% (Appendix A), and the peak was corresponded to the third peak of acetone extract of sample 458 (*E. mesomorpha*) in the same analytical condition (Appendix A).

### 2.4. MIC

The MIC values of *E. mesomorpha* extract against *S. aureus* and *E. faecium* were 64 and 16 μg/mL, respectively. The values coincided with the results of disk diffusion experiments. The purified divaricatic acid was found to be effective against *S. aureus* (32 μg/mL) and *E. faecium* (16 μg/mL) and against other Gram + bacteria, such as *B. subtilis*, *S. epidermidis*, *Str. mutans*, and *E. faecium* (5202) with MICs ranging from 7.0 to 64.0 μg/mL, whereas vancomycin had MICs ranging from 0.78 to 25.0 μg/mL (Table 3). Interestingly, the antibacterial activity of divaricatic acid was higher than that of vancomycin against *S. epidermidis* and *E. faecium* (5202). Furthermore, divaricatic acid was active against *C. albicans* (Table 3), whereas as vancomycin did not have antifungal properties [29], but slightly less effective than vancomycin against *S. aureus* (3A048; an MRSA). However, like vancomycin, divaricatic acid was not effective against Gram-bacteria. Divaricatic acid was more effective than cefotaxime, a broad-spectrum antibiotic, against *E. faecium* and the MRSA *S. aureus* (Table 3). MIC values of divaricatic acid against *B. subtilis* (7.0 μg/mL), *S. aureus* (64.0 μg/mL), and *C. albicans* (20.0 μg/mL) found in the present study were much lower than those (1.56, 1.56, 6.25 mg/mL, respectively) previously reported by Marijana et al. [25]. We attribute this discrepancy possibly to the types of strains tested, the culture conditions, the purity of the compound, and other unknown factors.

Structure of divaricatic acid, a depside, is quite different from those of well-known antibiotics. Usnic acid is a dibenzofuran having a structural unit similar to divaricatic acid and produced by various lichen species. It was reported that antibacterial activity of usnic acid was primarily caused by inhibition of DNA and RNA synthesis [30]. Based on the result, it might be suggested that divaricatic acid is related to the inhibition of nucleotide synthesis rather than cell-wall disruption. However, further study on the mechanism of divaricatic acid will be necessary. On the other hand, divaricatic acid showed a high degree of *Escherichia coli* RecA inhibition, relating to SOS response in bacteria [31]. The property might give an advantage in reduction of resistance induction. Toxicity of divaricatic acid to human cells was not reported, though gyrophoric acid, a depside (tridepside), showed no toxicity to human keratinocytes HaCaT cells [32]. The toxicity experiment should be needed in further study.

## 3. Materials and Methods

### 3.1. Chemicals and Media

Dimethyl sulfoxide (DMSO), ethyl acetate (EA), methanol (MeOH), chloroform, toluene, dioxane, dichloromethylene (DCM), acetic acid, and vancomycin were purchased from Sigma-Aldrich (St. Louis, MO, USA). Marine broth was obtained from MB Cell (Seoul, Korea). Nutrient broth, brain heart infusion (BHI), peptone, yeast extract, and agar were from Becton, Dickinson and Company (Sparks, MD, USA). Glucose was from Junsei (Tokyo, Japan) to prepare the GPYA (glucose-peptone-yeast extract-agar) medium.

### 3.2. Extracts of Lichen and Lichen-Forming Fungus Cultures

Acetone extracts of lichens (177 samples) and EA extracts of cultured lichen-forming fungi (LFF) (258 samples, 100 mg/mL) were obtained from the Korean Lichen Research Institute (KoLRI) in Sunchon National University, Korea. Samples were collected at various sites in Korea, China, Chile, Cameroon, Philippines, Romania, Vietnam, and the Arctic. Acetone extracts were dried and dissolved in DMSO (100 mg/mL).

### 3.3. Microbial Strains

*S. aureus* CCARM 3A048 (an MRSA), and *E. faecium* CCARM 5202, which were used for screening, were obtained from the Culture Collection of Antimicrobial Resistant Microbes (CCARM), Korea. To evaluate antimicrobial properties, we used five strains of Gram-positive bacteria (*Bacillus subtilis* KCTC 2189, *Micrococcus luteus* CCARM 0022, *S. aureus* CCARM 0027, *S. epidermidis* CCARM 3709, and *Streptococcus mutans* CCARM 0079), five strains of Gram-negative bacteria (*Escherichia coli* KCTC 2441, *Pseudomonas aeruginosa* CCARM 0057, *Klebsiella pneumoniae* CCARM 0015, *Salmonella typhimurium* CCARM 0240, and *Vibrio vulnificus* KCTC 2959), and *C. albicans* KCTC 27242 (a fungal strain).

### 3.4. Antibacterial Activity Assay

*S. aureus* CCARM 3A048 and *E. faecium* CCARM 5202 were grown at 37 °C for 24 h in BHI broth medium (calf brains 7.7 g, beef heart 9.8 g, protease peptone 10.0 g, dextrose 2.0 g, sodium chloride 5.0 g, disodium phosphate 2.5 g/L, pH 7.2–7.4) with shaking at 150 rpm. Cultures (0.1 mL) were then spread on BHI agar plates and paper disks (8 mm, Advantec, Toyo Roshi Kaisha. Ltd., Tokyo, Japan) sterilized with UV for 10 min, spotted with extracts (0.1 mL), and dried for 2–3 h, were placed on plates with spotted surfaces contacting the agar. For primary screening, aliquots of five extracts (1.0 mg each) were placed on a plate. Control of 10% EA or 10% DMSO, and vancomycin (20 or 40 ppm for *E. faecium* CCARM 5202 and *S. aureus* CCARM 3A048, respectively) were included. Plates were incubated at 37 °C for 12 h and the diameters of halos around the disks were observed or measured. Effective samples were selected by second screening (conducted in duplicate) using 0.5 mg of extract.

### 3.5. Analysis of Compounds in Extracts

Compounds in extracts with high antibacterial activity were analyzed at KoLRI by HPLC (LC-10AT, Shimadzu, Japan) equipped with a YMC-Pack ODS-A column using a solvent of methanol, water, and H_3_PO_4_ (80:20:1, *v*:*v*) as eluent [33], and putatively identified by comparing their retention times with a database, as previously described [34].

### 3.6. Isolation and Purification of Active Compounds and Structure Analysis

Compounds in the selected lichen (*E. mesomorpha*) extract were isolated by thin layer chromatography (TLC). The acetone extract was dissolved in 1 mL acetone in 1.5 mL EP tubes. For the preliminary study, this solution was spotted on Silica gel 60 F254 pre-coated plates (Merck Millipore, Darmstadt, Germany). The TLC plate was spotted with the sample and loaded in a twin trough chamber containing toluene, dioxane, and acetic acid (180:45:5, *v*/*v*), as previously described by Culberson [35]. Plate was removed from the chamber when the solvent front reached 15 cm from baseline and observed under 254 and 365 nm light. The plate was then sprayed with 10% aqueous sulfuric acid and heated for 5 min at 50 °C. For purification of the components, the lichen extract was separated by preparative TLC (PTLC) (silica gel 60 F254, 0.5 mm) using a toluene, dioxane, and acetic acid (90:25:4, *v*/*v*) mobile phase [24]. Spots were visualized at 254 nm and the major spot was removed. The silica gel obtained was then washed with MeOH and DCM (5:95, *v*/*v*), and the filtrate so obtained evaporated to dryness. Analysis of the compound was performed by a liquid chromatography–mass spectrometry (LC-MS) (LCMS-2020, Shimadzu, Kyoto, Japan) equipped with a reverse-phase column (Shim-pack VP-ODS/-C8/-Phenyl, 250 × 4.6 mm, Shimadzu). Elution was performed using combination of two solvents, that is, 1% formic acid in water (A) or acetonitrile (B). Linear gradient elution was performed using the following schedule: 5% B for 0–5 min; 5% to 30% B over 5–10 min; 30% B for 10–15min; 30 to 70% B over 15–20 min; and 70% B for 20–25 min. The structure of the compound was determined by ^1^H-NMR (JNM-ECZ400s/L1, JEOL, Tokyo, Japan).

### 3.7. Minimum Inhibitory Concentration (MIC)

Strains to be tested were grown at 37 °C for 24 h with shaking at 150 rpm. Nutrient agar was used for *B. subtilis* KCTC 2189 and *E. coli* KCTC 2441; marine agar for *V. vulnificus* KCTC 2959; and GPYA for *C. albicans* KCTC 27242. Other strains were grown in BHI medium. Cell densities were adjusted to an OD of 1.0 at 600 nm and diluted to 500 times. The isolated compound was dissolved in DMSO to produce a 10 mg/mL stock solution, then diluted to 256 μg/mL with BHI medium, then diluted serially twice from 1st to 10th well in a 96-well plate using 50 μL of BHI medium. Thereafter, 50 μL of diluted cells were added to 11th to 1st wells and incubated at 37 °C for 12 h. Each well contained 100 μL of liquid; 11th contained no compound or vancomycin, and 12th contained the medium only without the cell. MIC values were determined by comparing well turbidities with duplicate experiments and were presented as means ± standard errors.

## 4. Conclusions

In this study, divaricatic acid was isolated from the lichen *E. mesomorpha* and identified by LC-MS, ^1^H-, ^13^C- and DEPT-NMR. Divaricatic acid was found to be active against Gram + bacteria, such as *B. subtilis*, *S. epidermidis*, *S. mutans,* and *E. faecium,* and to show an antimicrobial activity against *C. albicans*. In addition, divaricatic acid was as effective as vancomycin against *S. aureus* (3A048; an MRSA). These results suggested that divaricatic acid may offer a means of treating MRSA infections.

## Figures and Tables

**Figure 1 molecules-23-03068-f001:**
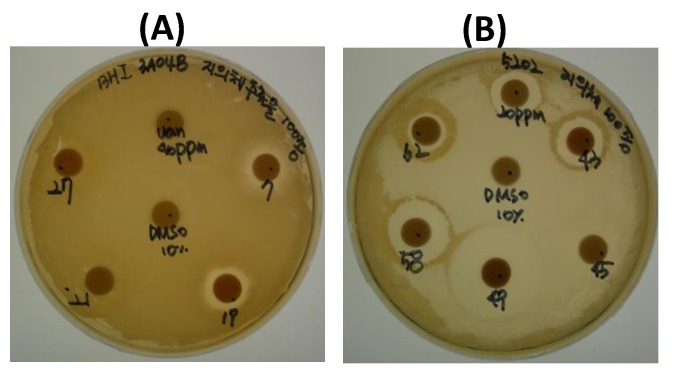
Antimicrobial activity as determined by the disk diffusion method. (**A**) *S. aureus*; (**B**) *E. faecium.*

**Figure 2 molecules-23-03068-f002:**
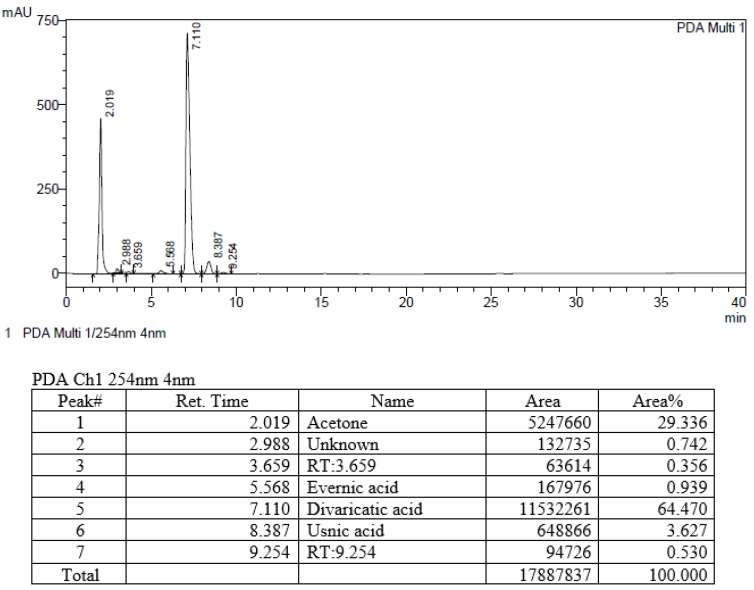
HPLC chromatogram and peak details in sample 458 (*Evernia mesomorpha*).

**Figure 3 molecules-23-03068-f003:**
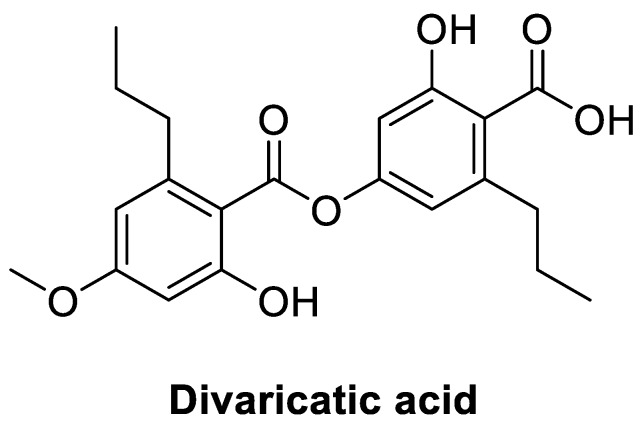
Structure of the compound identified as divaricatic acid.

**Table 1 molecules-23-03068-t001:** Antimicrobial activity assay results for lichen extracts as determined by the disk diffusion method *.

Sample No.	Lichen Name	*S. aureus* (3A048)	*E. faecium* (5202)
403	*Lobaria* sp.	-	9.0 ± 1.41
407	*Usnea* sp.	11.8 ± 0.35	23.0 ± 1.41
412	*Everniastrum* sp.	-	13.0 ± 1.41
419	*Everniastrum* sp.	12.0 ± 2.83	14.5 ± 0.71
421	.	-	11.8 ± 0.35
422	.	-	12.0 ± 2.83
431	*Rhizoplaca chrysoleuca*	11.8 ± 0.35	-
433	*Evernia divaricata*	-	13.0 ± 1.41
435	*Allocetraria ambigua*	-	9.8 ± 0.35
442	*Everniastrum nepalense*	-	10.5 ± 0.71
443	*Flavocetraria cucullata*	-	12.5 ± 0.71
445	*Thamnolia vermicularis*	-	11.0 ± 1.41
449	*Rhizoplaca chrysoleuca*	10.0 ± 0.00	23.8 ± 0.35
458	*Evernia mesomorpha*	9.0 ± 0.50	17.0 ± 1.41
462	*Ramalina* sp.	-	9.8 ± 0.35
466	*Usnea* sp.	9.5 ± 0.72	16.0 ± 2.83
470	*Ramalina* sp.	-	15.0 ± 1.41
471	*Usnea* sp.	-	15.0 ± 1.41
472	*Usnea* sp.	-	15.0 ± 1.41
473	*Niebla ceruchoides*	-	10.3 ± 0.35
493	*Ramalina* sp.	-	14.5 ± 0.71
504	*Cladonia* sp.	-	10.5 ± 0.71
511	*Usnea articulata*	-	12.5 ± 0.71
514	*Parmotrema* sp.	-	15.0 ± 2.83
517	*Ramalina* sp.	9.0 ± 0.50	17.0 ± 1.41
518	*Ramalina* sp.	-	16.3 ± 0.35
519	*Usnea* cf. *scabrida*	10.3 ± 0.35	14.5 ± 0.71
523	*Usnea* sp.	9.75 ± 0.35	13.5 ± 0.71
535	*Heterodermia* sp.	-	11.5 ± 0.71

* The values were obtained from duplicate experiments and were expressed as mm; 0.5 mg of lichen extract was applied to the disks. ., unidentified; -, not detected.

**Table 2 molecules-23-03068-t002:** Components in selected lichen extracts as determined by HPLC.

No.	Lichen Name	Country	HPLC
412	*Everniastrum* sp.	Vietnam	Galbinic acid, Atranorin, Chloroatranorin
419	*Everniastrum* sp.	Vietnam	Galbinic acid, Atranorin, Chloroatranorin
421	.	China	Unknown, Usnic acid
422	.	China	Galbinic acid, Usnic acid
435	*Allocetraria ambigua*	China	Usnic acid
442	*Everniastrum nepalense*	China	Galbinic acid, Usnic acid
458	*Evernia mesomorpha*	China	Unknown, Divaricatic acid, Usnic acid
473	*Niebla ceruchoides*	Chile	Usnic acid

., unidentified.

**Table 3 molecules-23-03068-t003:** MICs of divaricatic acid against bacteria and *Candida albicans **.

Strains		Divaricatic Acid	Vancomycin	Cefotaxime
		(μg/mL)	(μg/mL)	(μg/mL)
*Bacillus subtilis*	Gram +	7.0 ± 2.0	0.78 ± 0.0	0.5 ± 0.0
*Micrococcus luteus*		40.0 ± 16.0	25.0 ± 0.0	1.0 ± 0.0
*Staphylococcus epidermidis*		16.0 ± 0.0	25.0 ± 0.0	0.5 ± 0.0
*Streptococcus mutans*		32.0 ± 0.0	12.5 ± 0.0	0.5 ± 0.0
*Staphylococcus aureus* (0027)		64.0 ± 0.0	25.0 ± 0.0	64.0 ± 0.0
*Enterococcus faecium* (5202)		16.0 ± 0.0	25.0 ± 0.0	>256.0
*Escherichia coli*	Gram –	>256.0	>100.0	0.5 ± 0
*Pseudomonas aeruginosa*		128.0 ± 0.0	31.25 ± 12.5	32.0 ± 0
*Klebsiella pneumoniae*		>256.0	>100.0	>100.0
*Salmonella typhimurium*		>256.0	>100.0	0.5 ± 0
*Vibrio vulnificus*		>256.0	>100.0	8.0 ± 0
*Candida albicans*	Fungus	20.0 ± 8.0	>100.0	>256.0
*Staphylococcus aureus* (3A048)	MRSA	32.0 ± 0.0	25.0 ± 0.0	>256.0

* The values obtained from duplicate experiments are presented as means ± standard errors.

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
