# Peer review of "Antimicrobial Activity of Divaricatic Acid Isolated from the Lichen Evernia mesomorpha against Methicillin-Resistant Staphylococcus aureus"

_molecules, 2018, doi:10.3390/molecules23123068_

Round 1

Reviewer 1 Report

The comments are described in the attached file.

Author Response

We deeply appreciate for your kind review and valuable comments to our manuscript. We revised the manuscript according to the suggestions by Reviewers’ Comments. The revised portions were shown in red color.

1. E. mesomorpha was selected to study, because it had more abundant divaricatic acid. However, Usnea sp. showed the best antimicrobial activity in Table 1. Please explain why Usnea sp. wasn’t chosen to analyze?

As commented, Usnea sp. showed the best activity. After screening step, we checked the previous results about Usnea sp. from literature. The main component was usnic acid, the well-known compound. So, in view of its antimicrobial activity, novelty, and availability, E. mesomorpha producing divaricatic acid was selected for further study. Therefore, we added the sentences as below:

Usnea sp. showed the best antimicrobial activity, however, the major component was usnic acid; sample 458 showed less activity than Usnea sp., however, the major component was not well studied in its antimicrobial activity. In view of its antimicrobial activity, novelty, and availability, E. mesomorpha and divaricatic acid were selected for further study.

2. The HPLC chromatograms of 421 and 473 (Figure S3 and S7) are the same. Please confirm that the Figure S3 and S7 are correct?

Figure S7 was replaced with correct one. We appreciate your careful review.

3. The peaks of acetone and galbinic acid almost overlap in Figure S1, S2, S4 and S6. Please choose the better solvent system for analysis.

We putatively identified components of the extracts by comparing their retention times with a database as described in ‘Materials and methods’. The solvent was methanol, water, and H3PO4 (80:20:1, v:v). The database was constructed only with the solvent system. If further experiments are needed, then we use other solvent system. Galbinic acid was not our concerns, due to the published activity.

4. The retention time of divaricatic acid is 11.253 in Figure S12, but it is 7.110 in Figure 2. Please analyze the extract and standard samples under the same analytical condition.

HPLC experiments were performed under the same analytical condition, and the chromatogram of extract of sample 458 (Evernia mesomorpha) was added to Supplementary material as Figure S12 (A). The retention time of third peak was 11.407, similar to that of divaricatic acid, 11.253. The result was added as below:

Purity of the divaricatic acid by LC was 97.1% (Fig. S12B), and the peak was corresponded to the third peak of acetone extract of sample 458 (E. mesomorpha) in the same analytical condition (Fig. S12A).

5. In the 1H-NMR data of divaricatic acid, 6.38 (q, J = 2.9 Hz, 2H, H-3, H-5) should be change to 6.?? (d, J=?Hz, 1H, H-3) and 6.?? (d, J=?Hz, 1H, H-5). In addition, 0.95-1.02(m, 6H, H-3’’, H-3’’’) also should be change to 1.??(t, J-?Hz, 3H, H-3’’ or H-3’’’) and 0.9?(t, J=?Hz, 3H, H-3’’ or H-3’’’).

1H-NMR data of divaricatic acid was changed in Figure S9 and revised as below:

1H-NMR (400 MHz, CHLOROFORM-D) δ 11.35 (s, 1H, COOH), 6.75 (d, J = 2.1 Hz, 1H, H-5’), 6.64 (d, J = 2.3 Hz, 1H, H-3’) , 6.39 (d, J = 2.7 Hz, 1H, H-3, 6.38 (d, J = 2.7 Hz, 1H, H-5), 3.84 (s, 3H, OCH3), 2.93-3.02 (m, 4H, H-1’’, H-1’’’), 1.63-1.73 (m, 4H, 2xCH2, H-2’’, H-2’’’), 0.95(t, J=7.2Hz, 3H, H-3’’), 0.98(t, J=7.2Hz, 3H, H-3’’’)

6. Please explain why the chemical shift of C-3’(d111.56) and C-3(d98.98) are so different.

In shilding system, an electron donating group (EDG) or electron releasing group (ERG) (+I effect) is an atom or a functional group that donates a part of its electron density into a conjugated π system via resonance or inductive effects. In divaricatic acid, C-3 has EDG (-OMe and -OH) at two ortho positions. On the other hand, C-3 'has one EDG (-OH) and one EWG (-OOC) at the ortho position. As a result, C-3 position peak is located at more up field.

7. Please modify the carbon number as shown below.                                           

The number was modified as commented. We appreciate your careful review.

Reviewer 2 Report

These authors have isolated a library of natural products in an effort to identify some new antibiotic drugs. Overall, the methods and interpretation of data are sound and I believe it is of high interest. Thus, I recommend publication after minor revisions as noted.

Could the authors discuss more about the proposed mechanism of action for these compounds and the liklihood that they may or may be prone to induction of resistance?

Disk diffusion and MIC assays are good. But what about toxicity to human cells? Also, can the authors distinguish between bacteriostatic versus bactericidal activity? With the present data, it's not easy to differentiate these two possibilities.

Author Response

We deeply appreciate for your kind review and valuable comments to our manuscript. We revised the manuscript according to the suggestions by Reviewers’ Comments. The revised portions were shown in red color.

1. Could the authors discuss more about the proposed mechanism of action for these compounds and the liklihood that they may or may be prone to induction of resistance?

► It is difficult to propose the mechanism of action of divaricatic acid, since structure of divaricatic acid, a depside, is quite different from those of well-known antibiotics. We discussed the mechanism of divaricatic acid, based on the report that usnic acid, which is a dibenzofuran and produced by various lichens, causes inhibition of DNA and RNA synthesis. Usnic acid has a structural unit similar to divaricatic acid. The property might give an advantage in reduction of resistance induction. So, the discussions were added as below:

Structure of divaricatic acid, a depside, is quite different from those of well-known antibiotics. Usnic acid is a dibenzofuran having a structural unit similar to divaricatic acid and produced by various lichen species. It was reported that antibacterial activity of usnic acid was primarily caused by inhibition of DNA and RNA synthesis [30]. Based on the result, it might be suggested that divaricatic acid is related to the inhibition of nucleotide synthesis rather than cell-wall disruption. However, further study on the mechanism of divaricatic acid will be necessary. On the other hand, divaricatic acid showed a high degree of Escherichia coli RecA inhibition, relating to SOS response in bacteria [31]. The property might give an advantage in reduction of resistance induction.

2. Disk diffusion and MIC assays are good. But what about toxicity to human cells? Also, can the authors distinguish between bacteriostatic versus bactericidal activity? With the present data, it's not easy to differentiate these two possibilities.

► Toxicity of divaricatic acid to human cells was not reported, though it was reported that gyrophoric acid, a depside (tridepside), showed no toxicity to human keratinocytes HaCaT cells [Nguyen et al. 2013]. Unfortunately the experiment is not available in our laboratory at this time. It should be analyzed in further study and the point was mentioned in the text. As commented, it’s not possible to distinguish between bacteriostatic versus bactericidal activity with the present data. It can be differentiated by investigation of the mechanism. Based on the descriptions about usnic acid as above (i.e., DNA synthesis rather than cell-wall disruption), it might be predicted that divaricatic acid is bacteriostatic rather than bactericidal activity. However, further study about it will be needed. So, the discussions were added as below:

Toxicity of divaricatic acid to human cells was not reported, though gyrophoric acid, a depside (tridepside), showed no toxicity to human keratinocytes HaCaT cells [32]. The toxicity experiment should be needed in further study.

Reviewer 3 Report

Major revisions should be made, and the manuscript should be completed and/or modified as presented in attached file

Author Response

We deeply appreciate for your kind review and valuable comments to our manuscript. We revised the manuscript according to the suggestions by Reviewers’ Comments. The revised portions were shown in red color.

1. The authors should better use the term higher instead of greater (line 19, 106, 186). Vancomycin is an antibiotic used to treat bacterial infections caused by Gram-positive bacteria and does not have antifungal properties (T.S.N. Ku et al. Susceptibility of Candida albicans biofilms to azithromycin, tigecycline and vancomycin and the interaction between tigecycline and antifungals. International Journal of Antimicrobial Agents 36 (2010) 441 446.). Therefore the authors should rephrase the following: divaricatic acid was 5.0 times more active than vancomycin against Candida albicans / and to show greater antimicrobial activity against C. albicans than vancomycin (line 20-21, 107-108, 186-187).

We revised the term ‘greater’ to ‘higher’ at the three sites as commented. And, we revised the sentence properly as commented at three sites as below, including the reference suggested:

… and divaricatic acid was active against Candida albicans.

Furthermore, divaricatic acid was active against C. albicans (Table 3), whereas as vancomycin did not have antifungal properties [29],

… to show an antimicrobial activity against C. albicans.

2. Instead of vancomycin, the authors should have used another control compound with antifungal effect, and one active against G- bacteria for the evaluation.

► Cefotaxim, a broad-spectrum antibiotic, was added to the experiments. Discussion was added to the section as below:

Divaricatic acid was more effective than cefotaxime, a broad-spectrum antibiotic, against E. faecium and the MRSA S. aureus (Table 3).

3. The authors should rephrase the following: “ and produce numerous biologically active secondary metabolites with, for example, antiviral, antibiotic, antitumor, or allergenic properties” (lines 29-30).

► The part was revised to “… for example, antiviral, antibiotic, antitumor, or anti-allergenic properties”.

4. The authors should rephrase the following: “Usnic acid is probably the most investigated of these metabolites, and has antibiotic activity and have been used to develop pharmaceutics and cosmetics” (lines 32-33)

► The part was revised to “… the most investigated compound of these metabolites, and has antibiotic activity and has been used to develop pharmaceutics and cosmetics”

5. The authors should rephrase the following: “Enterococcus faecium is a Gram-positive commensal bacterium found in the human gastrointestinal tract, causes diseases, such as neonatal meningitis or endocarditis, and has recently emerged as a therapeutically challenging pathogen” (lines 44-46)

► The part was revised toEnterococcus faecium is a Gram-positive bacterium found in the human gastrointestinal tract, causes diseases, such as neonatal meningitis or endocarditis, and has recently emerged ... ”

6. The authors should rephrase the following: “Of the 177 acetone extracts of lichens, 6 and 28 samples were effective against S. aureus and E. faecium, respectively, and these samples were selected for further evaluation”. (lines 61-63)

► The part was revised to Of the 177 acetone extracts of lichens, 9 and 28 samples were effective against S. aureus and E. faecium, respectively, and 8 samples of them were effective against both strains. Total 29 samples were selected for further evaluation (Table 1).”

7. The authors should rephrase the following: “Antimicrobial activity testing of these 34 samples showed halo sizes for S. aureus were smaller than those for E. faecium (Fig. 1). In decreasing order, samples 419, 407, and 431 produced large halos for S. aureus, and similarly, samples 449, 407, 458, and 517 did so for E. faecium” (lines 65-67)

► The part was revised to In antimicrobial activity testing of these samples, halo sizes for S. aureus were smaller than those for E. faecium (Fig. 1). The tests were carried out with duplicate experiments. Diameter of halo was measured including disk (8 mm). Halo size order of the samples for S. aureus was 419 > 407 = 431, and those for E. faecium was 449 > 407 > 458 = 517 (Table 1).”

8. In order to obtain the results presented in table 1 and 3, how many determinations were made?

► For those experiments, two determinations were made. The points were added as notes at the end of Tables 1 and 3 as below:

*The values were obtained from duplicate experiments and were expressed as mm;

*The values obtained from duplicate experiments are presented as means ± standard errors.

9. The authors should better explain why they chose Evernia mesomorpha sample 458 - for further analysis? The results obtained by disc diffusion method show that on S. aureus the effect was not detected.

After screening step, we checked the previous results about Usnea sp. from literature. The main component was usnic acid, the well-known compound. So, in view of its antimicrobial activity, novelty, and availability, E. mesomorpha producing divaricatic acid was selected for further study. The reason for the selection of the lichen was added as below:

Usnea sp. showed the best antimicrobial activity, however, the major component was usnic acid; sample 458 showed less activity than Usnea sp., however, the major component was not well studied in its antimicrobial activity. In view of its antimicrobial activity, novelty, and availability, E. mesomorpha and divaricatic acid were selected for further study.

The experiments for the sample 458 on S. aureus were carried out again along with 449 and 517, showing larger than 17.0 mm of diameter against E. faecium. The results of the 449, 458, and 517 were 10.0 ± 0.00, 9.0 ± 0.50, and 9.0 ± 0.50 and added in Table 1.

10. What was the difference between samples 407, 466, 471, 472, 523? The lichen name is

the same. The authors should better explain the differences between analyzed samples.

The samples were identified in the genus level, Usnea. Species were not identified in the KoLRI, and were not available

11. The authors should also present the content (in mg/mL) of divaricatic acid in sample 458

(Evernia mesomorpha).

The extract was prepared by acetone extraction and evaporated to make solid compound at KoLRI. We used 100 mg/ml DMSO, as described in the text. It is difficult to measure the content in mg/ml of divaricatic acid in the extraction step. The ratio of divaricatic acid in the extract can be expected by the chromatogram; divaricatic acid was more abundant than usnic acid in E. mesomorpha extract as described in the text.

12. In tables 1 and 2, the authors should also present the names for samples 421 and 422.

 ► The samples were not identified for the names. So, the symbol was noted at the end of tables as below:

., unidentified.

13. The authors should use Italic style for m/z (line 91, 92, etc)

The words were changed to Italic style.

14. The authors should rephrase the following: “The 13C-NMR spectrum containing some 21 peaks (Fig. S10), and analysis of these peaks DEPT (distortionless enhancement by polarization transfer) (Fig. S11), resulted in its positive identification as divaricatic acid” (lines 95-97)

The sentence was revised to “Based on the twenty-one carbon peaks in 13C-NMR (Fig. S10) and protonated carbon shifts from DEPT (distortionless enhancement by polarization transfer) (Fig. S11), the compound was positively identified as divaricatic acid (Fig. 3).

15. For HPLC analysis, the authors should use another solvent system in order to separate the solvent (acetone) from compounds.

We putatively identified components of the extracts by comparing their retention times with a database as described in ‘Materials and methods’. The solvent was methanol, water, and H3PO4 (80:20:1, v:v). The database was constructed only with the solvent system. If further experiments are needed, then we use other solvent system.

16. The authors should compare the results obtained from disc diffusion method and MIC

Values.

► A sentence for the comparison was added as below:

The values coincided with the results of disk diffusion experiments.

17. The differences between present study and reference 25 cannot be due to the purity of compound: the values are 7 times higher and the purity was 97%. The authors should better explain.

► We revised the comment as below:

We attribute this discrepancy possibly to the types of strains tested, the culture conditions, the purity of the compound, and other unknown factors.

18. The authors presented very briefly their results. They should complete their manuscript

with discussion, as well as with Statistical Analysis section.

► We complete the manuscript with discussion, based on the other review comments as below:

Structure of divaricatic acid, a depside, is quite different from those of well-known antibiotics. Usnic acid is a dibenzofuran having a structural unit similar to divaricatic acid and produced by various lichen species. It was reported that antibacterial activity of usnic acid was primarily caused by inhibition of DNA and RNA synthesis [30]. Based on the result, it might be suggested that divaricatic acid is related to the inhibition of nucleotide synthesis rather than cell-wall disruption. However, further study on the mechanism of divaricatic acid will be necessary. On the other hand, divaricatic acid showed a high degree of Escherichia coli RecA inhibition, relating to SOS response in bacteria [31]. The property might give an advantage in reduction of resistance induction. Toxicity of divaricatic acid to human cells was not reported, though gyrophoric acid, a depside (tridepside), showed no toxicity to human keratinocytes HaCaT cells [32]. The toxicity experiment should be needed in further study.

► Description for statistical analytical section was added in the text subsections 2.1.1, 2.1.2, and 3.7 as bellows:

Based on halo sizes, samples tested were divided into three groups, i.e., low, moderate, and effective).

The tests were carried out with duplicate experiments.

… by comparing well turbidities with duplicate experiments and were presented as means ± standard errors

19. The authors should rephrase the following: “Samples were collected at various sites in Republic 126 of Korea, China, Chile, Cameroon, Philippines, Romania, and Vietnam and in the Arctic” (lines 126-127)

► The part was revised to “… at various sites in Republic of Korea, China, Chile, Cameroon, Philippines, Romania, Vietnam, and the Arctic.”

20. The authors should rephrase the following: “To evaluate antimicrobial properties we

used, five strains of Gram-positive” (line 132)

► The part was revised toTo evaluate antimicrobial properties, we used five strains of Gram-positive”.

Round 2

Reviewer 3 Report

The authors revised the manuscript according to the suggestions and provided specific answers, therefore the manuscript should be published in present form.